# NoFunEval: Funny How Code LMs Falter on Requirements Beyond Functional Correctness

**Manav Singhal, Tushar Aggarwal,**\* **Abhijeet Awasthi,**\* **Nagarajan Natarajan, Aditya Kanade**
manavsinghal157@gmail.com, {t-tuaggarwal,abawasthi,nagarajn,kanadeaditya}@microsoft.com
Microsoft Research India

## Abstract

Existing evaluation benchmarks of language models of code (code LMs) focus almost exclusively on whether the LMs can generate functionally-correct code. In real-world software engineering, developers think beyond functional correctness. They have requirements on "how" a functionality should be implemented to meet overall system design objectives like efficiency, security, and maintainability. They would also trust the code LMs more if the LMs demonstrate robust understanding of such requirements.

We propose a new benchmark NoFunEval to evaluate code LMs on *non-functional* requirements and simple classification instances for both functional and non-functional requirements. We propose a prompting method, *Coding Concepts* (*CoCo*), as a way for a developer to communicate the domain knowledge to the LMs. We conduct an extensive evaluation of 27 code LMs. Our finding is that LMs generally falter when tested on our benchmark, hinting at fundamental blindspots in their training setups. Surprisingly, even the classification accuracy on functional-correctness instances derived from the popular HumanEval benchmark is low, calling in question the depth of their comprehension and the source of their success in generating functionally-correct code in the first place. We release our benchmark and evaluation scripts publicly at https://aka.ms/NoFunEval.

## 1 Introduction

There has been dazzling progress in the development of newer and more capable language models (LMs) of code (Chen et al., 2021; Austin et al., 2021; Fried et al., 2022; Nijkamp et al., 2023; Li et al., 2023b; Wang et al., 2023; OpenAI, 2023b; Luo et al., 2023; Muennighoff et al., 2023). Simultaneously, the community has been actively designing benchmarks (Hendrycks et al., 2021; Chen et al., 2021; Austin et al., 2021; Puri et al., 2021; Li et al., 2022a; Liu et al., 2023b) with emphasis on *generating* code for a given problem specification of *what* functionality to achieve, e.g., writing a Python function to sort an array.

This is but a narrow slice of application of LMs in software engineering pipelines where the tasks are often not as straight-forward. Developers must consider the overall requirements of the system (e.g., an Android application) to which the code belongs. So, a problem instance in the real-world would be closer to editing Java code to *optimize for resource usage* on a low-memory Android device than to generating a functionally-correct sort. Such *non-functional requirements* guide the design decisions and constrain *how* the functionality may be realized (Landes & Studer, 1995), and play a central role in real-world software engineering (Chung et al., 2012). The premise of our work is that while satisfying functional requirements ("what" to implement) is necessary, it is not sufficient.

In this work, we forefront the above-identified significant gap in the current evaluation suites, and introduce a complementary benchmark NoFunEval in a first attempt to bridge the gap. We identify a set of *five broad non-functional requirements*: latency, resource utilization, runtime efficiency, maintainability, and security. We construct code-editing problem instances spanning these non-functional requirements and refer to them as NoFunEdit. As LMs

---

\*equal contribution

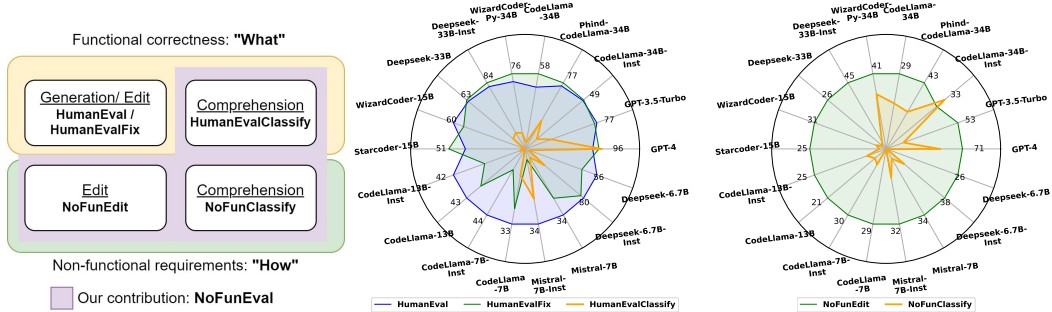

(a) Composition of NoFunEval    (b) Functional correctness    (c) Non-functional requirements

Figure 1: **(a)** NoFunEval contributes edit and comprehension tasks, NoFunEdit and NoFun-Classify, for non-functional requirements, and complements HumanEval and HumanEvalFix with a comprehension task HumanEvalClassify. **(b)–(c)**: Performance of LMs on NoFunEval, HumanEval, and HumanEvalFix benchmarks (metrics, full results in § 4). For consistency, in plot (c), we include only those instances with a binary evaluation oracle.

are finding increasing use in code generation and editing, it is imperative to test whether they have *robust comprehension of the requirements and code semantics*. We therefore propose classification instances for both functional-correctness, derived from the HumanEvalFix dataset (Muennighoff et al., 2023), and non-functional requirements, from NoFunEdit.

Figure 1(a) shows the three distinct subsets, NoFunEdit, NoFunClassify and HumanEvalClassify, of our NoFunEval benchmark and how they complement existing generation and edit benchmarks HumanEval (Chen et al., 2021) and HumanEvalFix (Muennighoff et al., 2023) focused on functional correctness. Our benchmark consists of 958 problem instances in multiple programming languages sourced from public repositories and existing datasets.

Two key challenges in our benchmark design are: (1) *how do we convey* notions like latency and efficiency that tend to be *relative* unlike functional correctness that is *absolute*? Simply describing the requirement in the prompt often fails because LMs may lack the necessary domain knowledge, unlike in the case of functional correctness where the problem description is usually sufficient. To this end, we design a new prompting strategy *Coding Concepts* (*CoCo*) which allows a developer to succinctly communicate actionable domain knowledge to LMs; (2) *how do we evaluate* the output of LMs? We employ a combination of functional (i/o specification) and non-functional (static analysis tools, execution time) oracles. In addition, we consider a DiffBLEU (Bairi et al., 2023), a metric defined on code diffs to capture the closeness of predicted edits with the ground-truth edits.

We present a comprehensive evaluation of 27 code LMs. A key takeaway of our work, besides the benchmark itself, is that existing code LMs (spanning different training strategies, instruction tuning paradigms, and model sizes) falter when we test them on requirements beyond functional correctness. This is highlighted in Figures 1(b)–1(c) by (1) their *surprisingly low performance on classification (yellow)* compared to generation (green) and editing (blue) tasks in both functional and non-functional requirements, and (2) the *generally low performance on tasks related to non-functional requirements* compared to relatively higher performances on tasks related to functional correctness.

**Contributions**: To the best of our knowledge, no prior work comprehensively evaluates code language models for multiple non-functional requirements in the context of code-editing and code-comprehension tasks. Our study is the first to observe that most code LMs typically fail to "discriminate" between buggy code and correct code, despite their ability to "generate" the correct fixes for the buggy code. In summary, we (1) identify the almost exclusive focus on functional correctness in existing benchmarks of code LMs; (2) prepare a benchmark to evaluate non-functional requirements and comprehension ability of code LMs; (3) extensively evaluate 27 code LMs and find there is much room to improve comprehension of requirements and code semantics; (4) release our benchmark and evaluation scripts at aka.ms/NoFunEval.

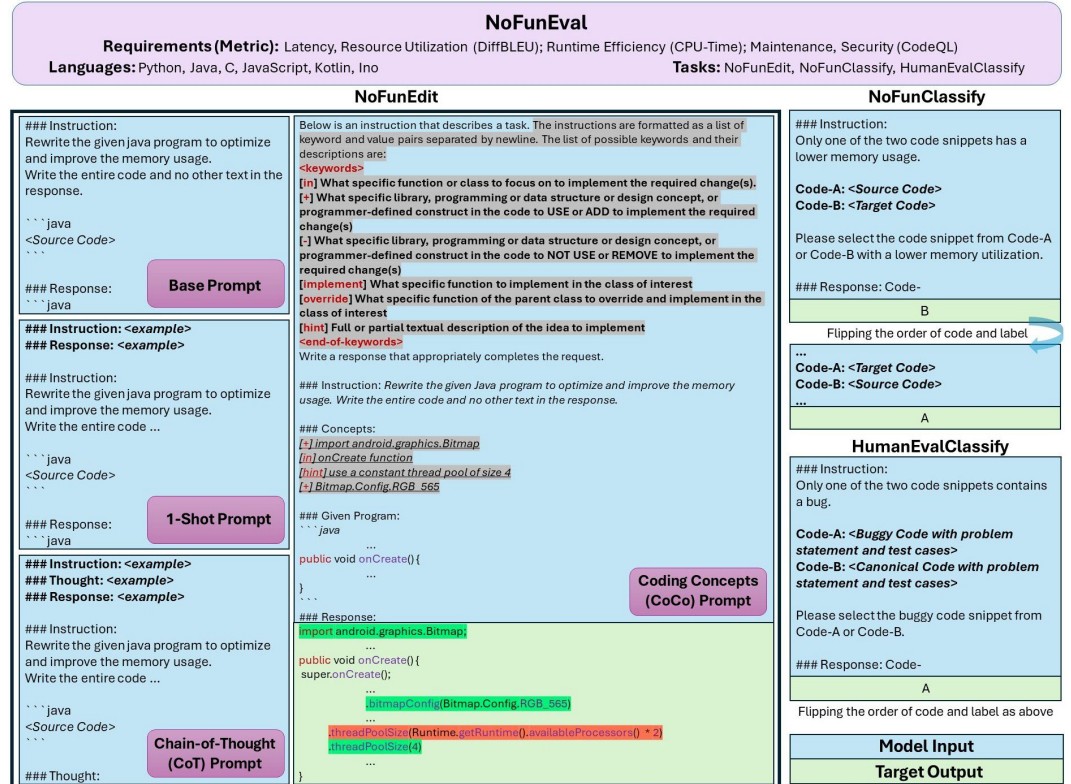

Figure 2: Overview of the NoFunEval benchmark. NoFunEval consists of three subtasks, NoFunEdit, NoFunClassify, HumanEvalClassify, spanning multiple programming languages. NoFunEdit (§ 2.1) involves editing a given source code as per a user-specified non-functional requirement (e.g., improving memory usage). We design four prompting techniques (§ 2.2) for eliciting LMs to perform the required editing, ranging from minimal task-related information ("Base") to guiding with high-level hints ("Coding Concepts"). NoFunClassify (§ 2.3) involves distinguishing between two code snippets based on a non-functional property (e.g., selecting the code with lower memory utilization). We construct it by reformulating problems in NoFunEdit. Similarly, we construct HumanEvalClassify (§ 2.4) by reformulating HumanEvalFix (Muennighoff et al., 2023), which involves distinguishing two code snippets based on their functional correctness (i.e., bug detection).

## 2 The NoFunEval Benchmark

Our NoFunEval benchmark (Figure 2) comprises one edit and two classification tasks. In this section, we describe these tasks, the design of the datasets, and the evaluation metrics (summarized in Tables A1 and A2 in Appendix A.1).

### 2.1 NoFunEdit

As shown in the LHS of Figure 2, each problem instance in NoFunEdit consists of an instruction specifying the non-functional requirement for code editing, a prompting strategy, and the source code that forms the input to the LM, along with the ground-truth code as the desired output. We consider five non-functional coding aspects: **(1) Latency**: Optimizing code for response times in applications, **(2) Resource Utilization**: Optimizing for resource utilization like memory, energy or bandwidth, **(3) Runtime Efficiency**: Improving algorithmic runtime complexity of code, **(4) Maintainability**: Enhancing code readability and style as per the best programming practices, and **(5) Security**: Resolving security vulnerabilities in code. Prior works have studied some of these aspects in isolation (§ 5). In contrast,

NoFunEdit attempts to unify these tasks under a general framework of editing the code to satisfy non-functional requirements. Below, we describe the design of NoFunEdit in detail.

**Latency and Resource Utilization**: Real-world software systems are often optimized for latency (e.g., network delays) and resource utilization (e.g., memory, energy) on edge devices. We turn to open-source Android applications that have commits optimizing latency and resource utilization. We derive such examples from Callan et al. (2022) and Moura et al. (2015), where they mine GitHub commits involving non-functional aspects of Android applications. The mining process involves keyword-based filters, manual selection, and learned classifiers, resulting in 931 examples covering latency (execution time and frame rate) and resource utilization (memory, bandwidth, and energy) properties. For our benchmark, we retain only commits from repositories with permissive licenses and targeting a single file. From the commits, we extract the pre-commit code as the input and the post-commit code as the target output. Based on the typical context size of 8192 tokens in many code-LMs, and the prompt lengths in our benchmark, we further filter out instances where the input source code length exceeds 3K tokens (using the StarCoder tokenizer). This results in a total of 114 examples, from which we manually discard 11 examples that do not conform to the commit message. Of the remaining 103, we reserve 5 examples for prompt construction (§ 2.2). The resulting 98 examples are then grouped based on their non-functional property – latency (47) and resource utilization (51). For evaluation, directly comparing latency or resource utilization is challenging, as examples vary in their target platforms (devices) and run time requirements (OS). So, we use the DiffBLEU score (Bairi et al., 2023) as our evaluation metric. DiffBLEU is designed to capture the similarity of *edits* made by the LM w.r.t. the target *edits*.

**Runtime Efficiency**: For assessing the ability to optimize code runtime, we derive examples from the Python test split of the PIE dataset (Madaan et al., 2023), which comprises 1K pairs of slow-code and fast-code for problems in the CodeNet challenge (Puri et al., 2021). We retain functionally-correct example pairs with at least two test cases (to ensure functional correctness). Further, we ensure (1) that each selected pair represents a unique CodeNet problem (to mitigate biases); (2) the target code is statistically significantly faster than the slower code in each selected pair, and (3) manually verify if the target edit is indeed a reasonable edit that can explain the observed speedup. This filtering results in 113 examples. We choose one example from the validation split of Madaan et al. (2023) for prompts (§ 2.2). For evaluation, we measure the average runtimes of the model-edited code and the target code over the test cases and over 25 repeated runs, and report the relative speedup. If the model-edited code is functionally incorrect or slower than the original code, we discard the model edits and assume the original code as output (i.e., a speedup of 1). We conduct experiments on Azure VM NC16 .

**Maintainability**: We derive examples corresponding to various maintainability-related issues from Sahu et al. (2024), which was in turn sourced from real-world git repositories (Raychev et al., 2016). We select 29 static checks that inspect code maintainability using CodeQL (Cod), a widely-used static analysis tool. We sample 5 examples per check (from thousands) from Sahu et al. (2024) for coverage, diversity, and economy. We ensure each selected code has token length less than 3K (as above). This results in a total of 145 instances, each with at least one maintainability-related issue flagged, and for which we manually write a valid target code. For evaluation, we test LM-edited outputs using the CodeQL tool. If CodeQL flags a warning related to the issue of interest, the LM output is considered a failure. On the other hand, absence of CodeQL warnings does not imply that the code is necessarily improved. LM could simply output empty code or delete offending lines of code, which suppresses warnings. So, to reward LM edits that are closer to the ground-truth edits, we weigh the binary CodeQL success/failure scores with the continuous DiffBLEU scores. We denote this metric by DiffBLEU$\times$CodeQL.

**Security**: For evaluating an LM's ability to fix security vulnerabilities, we repurpose the Pearce et al. (2022) dataset which covers 18 out of the top 25 Common Weakness Errors (CWE, 2021) scenarios. We use upto 2 generations from GitHub Copilot per CWE in the dataset with at least one security issue flagged by CodeQL. This results in total 41 examples across 13 CWEs. We manually write the reference code that addresses the flagged vulnerabilities. For evaluation, we use the DiffBLEU$\times$CodeQL metric as above.

Overall, the raw datasets from where we derive NoFunEdit have many more examples. However, to ensure the reliability of the evaluation, we only retained the examples that were strictly consistent with the non-functional requirements. Further, we prioritized inclusion of examples that are diverse and cover more types of non-functional requirements than having a greater number of similar examples.

**Use of DiffBLEU for Evaluation**: Our benchmark partly derives examples from files of end-to-end applications designed for diverse platforms like mobile devices, web applications, etc. Thus, setting up execution environments for such tasks is not scalable and expensive to run for the benchmark users. We therefore use the DiffBLEU (Bairi et al., 2023) metric, designed to specifically compare the "edits" generated by the model with the ground truth edits. DiffBLEU addresses the limitations of prior surface-level metrics like Code-BLEU (Ren et al., 2020) that do not explicitly focus on the edits made to the input file. We also observe a high correlation between DiffBLEU and execution-based metrics like accuracy as well as static-analysis based CodeQL scores (§ 4.5). Thus, DiffBLEU serves as a reasonable light-weight alternative to heavy-weight execution or static-analysis based oracles. Moreover, oracles can also be imperfect. For instance, test-cases or the CodeQL static checks though expert-designed, may not cover all the corner cases. Thus, we augment our oracles for Security and Maintainability by combining them with DiffBLEU scores, as described earlier.

## 2.2 Crafting LM Prompts for NoFunEdit

We design four types of prompts to elicit editing abilities in code-LMs (Figure 2). These vary from a minimalist specification of task requirements ("Base") to more comprehensive specification leveraging domain expertise ("Coding Concepts") as described below. Examples of all the prompt templates are given in Appendix A.2.

**Base Prompt**: For each non-functional requirement in NoFunEdit, we write a simple instruction that conveys the requirement at a high level. The (Base) example in Figure 2 shows how we specify the requirement of optimizing the resource (memory) utilization. Depending on the problem instance, resource utilization prompt specifies one of memory, bandwidth, or energy. Similarly, the prompt for improving latency uses the keywords frame-rate or execution time; and for runtime efficiency, execution time. The prompts for maintainability and security requirements utilize the title of CodeQL warnings flagged for the input code or the common weakness enumeration (CWE) respectively.

**1-Shot Prompt**: We expand on the Base prompt above, which is zero-shot, to include an example that shows how to implement the desired non-functional requirement. In particular, we give a pair of the original source code and the edited source code, as illustrated in the 1-Shot prompt of Figure 2. Considering very limited data available for each non-functional requirement, limited context lengths supported by various LMs, and each example containing code from entire file, we do not explore multi-shot prompts.

**Chain-of-Thought (CoT) Prompt**: Combining reasoning with few-shot examples via chain-of-thought has led to improved results across many tasks (Wei et al., 2022). This is particularly appealing for code rewrite tasks that require multiple levels of reasoning – understanding the implementation of the functionality (i.e., the *what*); the nature of the issue (e.g., memory leak); the source of the issue (i.e., localization); *how* the issue can be tackled (i.e., the exact code edit necessary). Considering these requirements, we manually augment each example in 1-Shot prompts with explanations (thoughts) eliciting such reasoning steps. As shown in the CoT prompt in Figure 2, the LM first generates a thought by attending to the thought-augmented example in the prompt, and then outputs its response conditioned on the reasoning steps in the generated thought.

**Coding Concepts (CoCo) Prompt**: CoT prompts rely on the LM's ability to generate reasoning steps for the required edits. This could potentially lead to poor judgements in terms of code localization and resolution, depending on the LM's generative abilities as well as its domain knowledge about the non-functional requirement. Often, developers have some idea of what they expect in the edits and can provide the domain knowledge they possess with respect to their codebase, such as what libraries to import for optimizing the code, or which part of code requires the edits, etc. To this end, we propose a simple and fairly general

prompting strategy – *Coding Concepts* (*CoCo*), that gives the LM hints on the programming concepts to use for the task. As shown in the middle pane of Figure 2, we first provide a legend of concepts and their descriptions. Then, for the problem instance, we give candidate values for the applicable concepts, serving as directions to the LM for *how* to implement the edits. The LM has to figure out how to compose the concepts to accomplish the desired task. To ensure high quality, we manually create the CoCo prompts for each problem instance, while also ensuring that hints in these prompts do not reveal the actual edits.

## 2.3 NoFunClassify

To study comprehension of non-functional requirements in code LMs, we repurpose the NoFunEdit task into a code-classification task called NoFunClassify; it involves distinguishing between two code snippets based on a non-functional property (e.g., selecting the code snippet with lower memory usage). The upper right of Figure 2 shows an example. We use standard accuracy as the evaluation metric for this task. However, to be agnostic to the ordering of code snippets in the prompt, we prompt the code LM using both the orderings separately. An LM is considered correct on the example only if *both* the orderings result in correct outputs. While NoFunEdit tests for code editing abilities, NoFunClassify tests only for code comprehension; thus, one would anticipate NoFunClassify to be easier than NoFunEdit.

## 2.4 HumanEvalClassify

Similar to NoFunClassify, we design HumanEvalClassify, but to test the comprehension of functional correctness in code LMs. We construct HumanEvalClassify by repurposing the Python split of HumanEvalFixDocs dataset from Muennighoff et al. (2023), which we refer to simply as HumanEvalFix. HumanEvalFix contains functionally incorrect and correct pairs of code, where the incorrect code is obtained by introducing synthetic bugs in the original examples of HumanEval dataset. To convert HumanEvalFix into a code comprehension task, we prompt LMs to select the incorrect code from the two code snippets (as in the bottom right of Figure 2). We use the same evaluation methodology as for NoFunClassify (§ 2.3).

## 3 Experimental Details

We evaluate multiple open and closed-weight code LMs: GPT-4 (OpenAI, 2023b), GPT-3.5-Turbo (OpenAI, 2023a), WizardCoder (Luo et al., 2023), StarCoder (Li et al., 2023b), CodeLlama (Roziere et al., 2023), Mistral (Jiang et al., 2023), DeepSeekCoder (Guo et al., 2024), CodeGemma (CodeGemma, 2024), and Llama3 (Dubey et al., 2024) model families. The open-weight models were downloaded from Huggingface; the model sizes vary from 1B to 70B parameters. We cover a total of 27 LMs in our study.

**Generating LM Outputs**: To decode LM outputs for NoFunEdit and HumanEvalFix, we use two sampling mechanisms: (1) sample 20 generations (per problem instance) with a temperature of 0.8, and (2) greedy sampling with temperature 0. We utilize top-$k$ (Fan et al., 2018) and nucleus sampling (Holtzman et al., 2019), with the default values of $p = 0.95$ and $k = 50$. All the LMs in our evaluation support context size of 8192 tokens. For the instances corresponding to runtime efficiency and security, which are of relatively shorter length, we restrict the maximum number of sampled tokens to 1200, as in Madaan et al. (2023), and to 1500 for CoT prompting to accommodate thoughts. For NoFunClassify and HumanEvalClassify, we do greedy sampling to generate output labels. We utilize the vLLM library (Kwon et al., 2023) for generating LM outputs for tasks in NoFunEval. For HumanEvalFix, we use BigCode's evaluation harness (Ben Allal et al., 2022).

**Evaluating LM Outputs**: We design evaluation metrics specific to each non-functional requirement as detailed in Section 2 and summarized in Table A1. For NoFunEdit, since we sample $n = 20$ candidate outputs per input, we report expected scores using the score@$k, n$ function (Agrawal et al., 2023), a generalization of the pass@$k, n$ function (Chen et al., 2021), that accounts for continuous metrics like DiffBLEU or average speed-up, in addition to discrete metrics. We primarily use score@1, 20 for reporting our observations in Section 4. Similarly, for HumanEvalFix, we report pass@1, 20 scores. For NoFunClassify and HumanEvalClassify, we report classification accuracy.

| Models | Latency | | Resource Util. | | Runtime Efficiency | | Maintainability | | Security | | HumanEvalFix |
|---|---|---|---|---|---|---|---|---|---|---|---|
| | Min | Max | Min | Max | Min | Max | Min | Max | Min | Max | Base Prompt |
| GPT-3.5-Turbo | 12.2 1S | 31.3 CoCo | 10.8 B | **29.9 CoCo** | 1.302 B | 1.774 CoCo | 21.0 B | 40.3 CoCo | 39.3 B | 55.6 1S | 72.3 |
| GPT-4 | **14.2 B** | 35.9 CoCo | 10.7 B | 26.3 CoCo | 1.303 B | **2.380 CoCo** | **33.2 B** | **51.1 CoCo** | **41.5 B** | 64.3 CoT/1S | **90.2** |
| StarCoder-1B | 1.9 1S | 4.5 CoCo | 1.3 1S | 4.8 CoCo | 1.004 1S | 1.009 CoT | 1.3 B | 2.2 CoT | 1.3 B | 14.4 1S | 7.7 |
| WizardCoder-1B | 0.2 CoT | 5.6 CoCo | 0.9 CoT | 3.9 CoCo | 1.002 1S | 1.008 CoCo | 1.7 CoT | 2.9 CoCo | 7.7 B | 26.7 1S | 19.6 |
| StarCoder-15.5B | 3.4 1S | 7.6 CoT/B | 5.0 CoCo | 8.0 CoT | 1.033 B | 1.056 CoCo | 4.5 B | 6.1 CoT | 16.5 B | 47.8 1S | 40.9 |
| WizardCoder-15.5B | 3.9 1S | 16.1 CoCo | 5.7 1S | 15.3 CoCo | 1.031 B | 1.183 CoCo | 6.8 B | 13.0 CoCo | 25.6 B | 51.7 1S | 51.6 |
| Mistral-7B | 4.4 1S | 12.0 CoCo | 4.7 1S | 9.3 CoCo | 1.010 1S | 1.132 CoCo | 4.6 B | 8.6 CoCo | 21.8 B | 42.5 1S | 28.3 |
| Mistral-7B-Inst | 6.6 1S | 12.7 CoCo | 6.0 1S | 9.5 CoCo | 1.011 B | 1.118 CoCo | 4.6 B | 8.0 CoCo | 23.0 B | 41.6 1S | 10.9 |
| CodeLlama-7B | 2.2 1S | 10.1 CoCo | 2.8 1S | 7.6 CoCo | 1.020 1S | 1.079 CoCo | 2.7 B | 6.6 CoCo | 13.2 B | 46.1 1S | 30.0 |
| CodeLlama-7B-Inst | 3.7 1S | 12.0 CoCo | 4.3 1S | 9.1 CoCo | 1.029 1S | 1.103 CoCo | 4.8 B | 12.3 CoCo | 19.3 B | 45.9 1S | 20.6 |
| CodeLlama-13B | 3.9 1S | 9.5 CoCo | 3.3 B | 8.5 CoCo | 1.051 B | 1.224 CoCo | 3.7 B | 7.2 CoCo | 14.6 B | 46.6 1S | 33.1 |
| CodeLlama-13B-Inst | 3.6 1S | 13.9 CoCo | 4.0 B | 11.6 CoCo | 1.037 1S | 1.259 CoCo | 5.5 B | 13.7 CoCo | 22.7 B | 48.8 CoT | 30.5 |
| CodeLlama-34B | 1.1 1S | 13.3 CoCo | 3.3 B | 8.2 CoCo | 1.064 B | 1.512 CoCo | 7.8 B | 15.8 CoCo | 23.2 B | 52.7 1S | 55.9 |
| CodeLlama-34B-Inst | 2.4 1S | 20.2 CoCo | 3.9 B | 11.3 CoCo | 1.052 B | 1.510 CoCo | 9.7 1S | 23.4 CoCo | 24.5 B | 54.7 1S | 47.6 |
| Phind-CodeLlama-34B | 4.7 1S | 29.2 CoCo | 5.0 1S | 21.5 CoCo | 1.148 B | 2.155 CoCo | 15.7 CoT | 38.8 CoCo | 33.0 B | 59.1 1S | 77.3 |
| WizardCoder-Py-34B | 12.7 CoT | 28.9 CoCo | **11.0 1S** | 23.8 CoCo | 1.076 B | 1.421 CoCo | 11.2 B | 19.8 CoCo | 31.5 B | 45.5 CoCo | 75.6 |
| Llama-3-8B-Inst | 0.1 1S | 6.3 CoCo | 0.5 1S | 4.4 CoCo | 1.062 CoCo | 1.128 B | 7.0 B | 11.7 CoCo | 22.9 B | 53.6 CoT | 41.5 |
| Llama-3-70B-Inst | 13.2 B | 34.6 CoCo | 10.7 1S | 27.6 CoCo | 1.172 B | 2.335 CoCo | 27.1 B | 44.4 CoCo | 37.2 B | 57.9 1S | 81.7 |
| CodeGemma-2B | 0.0 CoT | 7.1 B | 2.3 CoT | 7.6 1S | 1.002 1S | 1.004 CoCo/B | 2.2 B | 3.3 CoCo | 3.9 B | 47.0 CoT | 22.7 |
| CodeGemma-7B | 0.5 CoT | 9.1 CoCo | 1.3 1S | 6.1 CoCo | 1.006 CoT | 1.115 CoCo | 5.6 B | 12.0 CoCo | 28.5 B | 54.3 1S | 11.7 |
| CodeGemma-7B-Inst | 3.6 CoT | 21.0 CoCo | 2.9 CoT | 12.8 B | 1.031 B | 1.642 CoCo | 6.5 CoT | 21.7 CoCo | 42.4 B | 67.2 1S | 72.4 |
| DeepSeekCoder-1.3B | 2.1 1S | 4.7 CoCo | 3.0 1S | 3.9 CoCo | 1.007 B | 1.046 CoCo | 1.2 B | 3.3 CoT | 4.4 B | 24.2 CoT | 16.4 |
| DeepSeekCoder-1.3B-Inst | 7.4 1S | 12.0 CoCo | 5.8 B | 6.6 CoCo | 1.064 B | 1.184 CoCo | 3.7 CoT | 8.6 CoCo | 20.0 B | 32.7 CoCo | 48.9 |
| DeepSeekCoder-6.7B | 3.0 1S | 13.8 CoCo | 4.5 1S | 14.6 CoCo | 1.150 B | 1.404 CoCo | 5.1 1S | 12.5 CoCo | 20.2 B | 51.3 1S | 45.4 |
| DeepSeekCoder-6.7B-Inst | 8.6 1S | 22.0 CoCo | 7.5 1S | 19.6 CoCo | 1.413 CoT | 1.810 CoCo | 16.1 CoT | 29.7 CoCo | 32.9 B | 53.7 CoT | 73.3 |
| DeepSeekCoder-33B | 3.6 1S | 19.7 CoCo | 5.2 1S | 16.2 CoCo | 1.321 B | 1.524 CoCo | 10.6 B | 19.7 CoCo | 29.7 B | 52.4 1S | 61.6 |
| DeepSeekCoder-33B-Inst | 10.7 1S | 28.7 CoCo | 8.7 1S | 20.8 CoCo | **1.548 B** | 2.269 CoCo | 18.7 B | 32.2 CoCo | 29.5 B | 47.3 1S | 81.0 |
| Ground-truth Score | 100 | | 100 | | 3.7 | | 100 | | 100 | | 100 |

Table 1: Performance of code LMs on the NoFunEdit dataset by different non-functional requirements (§ 4.1, § 4.2). LMs from the same family or sharing the same base model are grouped together. For brevity, we only report the performance of the worst (Min) and the best (Max) performing prompt, with prompt type in the superscript abbreviated as Base (B), 1-Shot (1S), Chain-of-Thought (CoT), Coding Concepts (CoCo). The numbers correspond to the metrics discussed in Section 2 (higher is better; **highest** in bold; second-highest underlined). We present results for all the prompts in Appendix, Figure A.4. Additionally, we report results on HumanEvalFix for studying its difficulty relative to NoFunEdit.

# 4 Evaluation Results

## 4.1 Summary of Results: Two Key Takeaways

Tables 1 and 2 provide performance of all the LMs over code-editing (NoFunEdit) and code-classification tasks (NoFunClassify, HumanEvalClassify) respectively. We also report the performance of these LMs on HumanEvalFix for reference in Table 1. Overall, we find:

**(1) Code LMs struggle to edit code for satisfying non-functional requirements** (inferred from Table 1). This is evidenced by the gap between the scores received by them compared to the ground-truth score (the last row in Table 1). This suggests that the LMs have weak understanding of the non-functional requirements of code. Note that since the metrics vary as per the requirements, we cannot compare performances across requirements directly.
**(2) Code LMs fail to sufficiently comprehend code they can otherwise synthesize or edit** (inferred from Table 2). The accuracy of the LMs over NoFunClassify ranges from 0–42% and 0–95.7% for HumanEvalClassify, continuing the trend that non-functional requirements pose a challenge. GPT-4, the best performing model overall, performs much worse on three of the five non-functional tasks; on the other hand, it achieves 95.7% classification accuracy over functional tasks (i.e., HumanEvalClassify). To our surprise, we find that given an incorrect and a corresponding correct code snippet, many code LMs fail to distinguish between the two, but can successfully edit and fix the incorrect code snippet. For instance, DeepSeekCoder-33B-Inst fixes bugs in HumanEvalFix 81% of times (Table 1), but can distinguish between the incorrect and the correct code only 20.7% of times in HumanEvalClassify (Table 2). We observe similar trends across the LMs except GPT-4 (Figure A.10).

## 4.2 How Well do LMs Perform on NoFunEdit?

**Larger instruction-tuned models and CoCo prompts offer superior performance.** First, in accordance with the scaling laws (Kaplan et al., 2020; Hoffmann et al., 2022), we observe that larger models are consistently better than their smaller variants. Second, our proposed CoCo prompt is often the highest scoring prompting strategy (99 out of $27 \times 5 = 135$ cases). Figure A.7 in Appendix A.6 presents an example where GPT-4 utilizes hints in the CoCo prompt to arrive at the correct output, while all other prompts including CoT lead to undesired code edits. Third, instruction-tuning in general helps improve the model performance across all the prompts. For instance, compare the scores of StarCoder-15.5B (row 5) with that of its instruction-tuned version WizardCoder-15.5B (row 6). We also note that CoCo prompting strategy may lead to much larger improvements in instruction-tuned models compared to their base-variants — owing to the ability of the former models to follow the instructions encoding the domain knowledge in the CoCo prompt. For instance, on four out of five requirements, CoCo offers greater improvements over the Base prompt for DeepSeekCoder-33B-Inst, when compared to DeepSeekCoder-33B.

**Zero-shot base prompts outperform 1-shot prompts in several cases.** Initially, we anticipated higher performance using 1-Shot prompts given the in-context learning ability of LMs. Counter-intuitively, zero-shot base prompts offer superior results compared to 1-Shot prompts in several cases. A notable exception is the Security task, where 1-Shot prompts usually emerge the winner. This is not surprising given that there are at most one or two test instances per security vulnerability, and the 1-Shot example (derived from official CWE or CodeQL web pages) often captures the nature of the required edits. For the other non-functional requirements, while we could improve the choice of the example we pick for 1-Shot with additional efforts, we hypothesize that conditioning on 1-Shot examples may introduce unintended biases thereby restricting LMs to generalize beyond the provided example. Figure A.8 in Appendix A.6 provides a supporting anecdote. We expect multi-shot prompts with diverse examples to overcome this limitation. However, as discussed in Section 2.2, we could not explore multi-shot prompts due to limited data, and limited context lengths in LMs to support multiple file-sized prompts.

**Non-functional improvements may come at the cost of functional correctness.** For Runtime Efficiency tasks, we observe that LMs like GPT-4, GPT-3.5-Turbo, Phind-CodeLlama-34B, and Llama-3-70B-Inst yield code with significant runtime improvements. However, we also find that they often make edits that compromise on functional correctness while they improve the runtime (recall that in such cases, we simply ignore the suggested edits, and retain the input program as mentioned in Section 2.1). Figure A.6 in Appendix A.5 shows one such output obtained from GPT-4. This observation again points to the lack of fundamental understanding, e.g., the non-negotiable nature of functional-correctness while aiming for non-functional improvements, in code LMs. Appendix A.5 presents more detailed observations comparing run-time improvements with drops in execution accuracy (functional correctness).

## 4.3 How Well do LMs Perform on Classification Tasks?

Table 2 shows the results for classification tasks NoFunClassify (§ 2.3) and HumanEvalClassify (§ 2.4). We discover some counter-intuitive trends: **(1) No model is consistently the best across all the tasks**. For instance, GPT-4 is the best model only for Maintainability, Security, and Bug Detection tasks, and Llama-3-70B-Inst considerably outperforms GPT-4 on the remaining tasks of identifying programs with better Latency (+15.8%), Resource Utilization (+16.8%), and RunTime Efficiency (39.0%). **(2) Larger models or instruction-tuned variants may not be better than the corresponding smaller or base variants**. For instance, CodeLlama-13B outperforms CodeLlama-34B by 14.3% on the task of identifying code snippets with lower latency, and DeepSeekCoder-6.7B-Inst outperforms DeepSeekCoder-33B-Inst by 5.5% on HumanEvalClassify. Comparing between base models and their instruction-tuned variants, we notice that CodeLlama-34B outperforms CodeLlama-34B-Inst by 16.7% on identifying code snippets with lower resource utilization. Similarly, DeepSeekCoder-33B outperforms its instruct version by 22.4% on Latency examples. Worse performance of instruct models could be attributed to the lack of task diversity in instruction-tuning datasets.

| Models | Latency | Resource Util. | RunTime Efficiency | Maintain-ability | Security | NoFun-Classify | HumanEv-alClassify |
|---|---|---|---|---|---|---|---|
| GPT-3.5-Turbo | 22.4 | 9.3 | 8.0 | 12.4 | 26.8 | 13.4 | 28.0 |
| GPT-4 | 20.4 | 18.5 | 15.9 | **63.4** | **92.7** | **42.0** | **95.7** |
| StarCoder-1B | 0.0 | 0.0 | 0.8 | 0.0 | 0.0 | 0.2 | 0.0 |
| WizardCoder-1B | 0.0 | 0.0 | 1.7 | 0.0 | 0.0 | 0.5 | 0.0 |
| StarCoder-15.5B | 2 | 1.9 | 5.9 | 0.0 | 0.0 | 2.2 | 2.4 |
| WizardCoder-15.5B | 20.4 | 5.6 | 0.8 | 8.3 | 0.0 | 6.4 | 1.8 |
| Mistral-7B | 0.0 | 11.1 | 0.0 | 8.3 | 7.3 | 5.2 | 3.0 |
| Mistral-7B-Inst | 8.2 | 0.0 | 17.8 | 9 | 9.8 | 10.3 | 23.8 |
| CodeLlama-7B | 0.0 | 0.0 | 0.0 | 0.0 | 0.0 | 0.0 | 10.4 |
| CodeLlama-7B-Inst | 4.1 | 3.7 | 11 | 2.8 | 12.2 | 6.4 | 0.0 |
| CodeLlama-13B | 20.4 | 14.8 | 0.8 | 6.9 | 2.4 | 7.3 | 1.8 |
| CodeLlama-13B-Inst | 2 | 0.0 | 2.5 | 6.2 | 19.5 | 5.2 | 4.9 |
| CodeLlama-34B | 6.1 | 20.4 | 39 | 4.1 | 4.9 | 16.4 | 6.1 |
| CodeLlama-34B-Inst | 4.1 | 3.7 | 55.1 | 10.3 | 48.8 | 25.4 | 8.5 |
| Phind-CodeLlama-34B | 18.4 | 16.7 | 22.9 | 18.6 | 46.3 | 22.4 | 34.1 |
| WizardCoder-Py-34B | 10.2 | 9.3 | 47.5 | 15.9 | 31.7 | 25.0 | 15.9 |
| Llama-3-8B-Inst | 0.0 | 0.0 | 8.0 | 2.1 | 7.3 | 3.8 | 4.3 |
| Llama-3-70B-Inst | **36.2** | **35.3** | **54.9** | 16.6 | 29.3 | 33.5 | 69.5 |
| CodeGemma-2B | 4.3 | 5.9 | 13.3 | 4.1 | 12.2 | 7.8 | 12.8 |
| CodeGemma-7B | 0.0 | 0.0 | 0.0 | 0.0 | 0.0 | 0.0 | 6.7 |
| CodeGemma-7B-Inst | 10.6 | 33.3 | 26.5 | 11 | 22 | 19.4 | 54.3 |
| DeepSeekCoder-1.3B | 0.0 | 0.0 | 5.9 | 0.0 | 0.0 | 1.7 | 2.4 |
| DeepSeekCoder-1.3B-Inst | 14.3 | 5.6 | 16.9 | 0.0 | 2.4 | 7.5 | 3.0 |
| DeepSeekCoder-6.7B | 0.0 | 0.0 | 0.8 | 0.0 | 0.0 | 0.2 | 12.2 |
| DeepSeekCoder-6.7B-Inst | 2 | 3.7 | 14.4 | 13.1 | 0.0 | 9.6 | 26.2 |
| DeepSeekCoder-33B | 30.6 | 14.8 | 0.8 | 2.1 | 2.4 | 6.8 | 12.8 |
| DeepSeekCoder-33B-Inst | 8.2 | 3.7 | 2.5 | 7.6 | 2.4 | 5.2 | 20.7 |
| Average | 8.8 | 6.8 | 11.8 | 8.4 | 13.5 | 9.7 | 13.1 |
| Maximum | 30.6 | 20.4 | 55.1 | 63.4 | 92.7 | 42.0 | 95.7 |

Table 2: Accuracy of LMs (**highest**; second-highest) on the 5 non-functional requirements, full NoFunClassify (micro-averaged over the 5 requirements), and HumanEvalClassify.

## 4.4 How do Comprehension Abilities Compare with Edit Abilities?

From Figures 1(b) and 1(c), we find that **LMs are relatively more accurate in code editing compared to the corresponding code comprehension tasks.** Notably, this observation holds not only for non-functional requirements but also for functional correctness. For instance, DeepSeekCoder-33B-Inst can correctly edit 81% of buggy code in the HumanEvalFix dataset, but it can discriminate between a buggy and the corresponding correct code only 20.7% of the times. Figure A.9 in Appendix A.6 shows an example. In Figure A.10 of Appendix A.6, we present a performance breakdown for all LMs on classification (HumanEvalClassify) and the corresponding edit instances (HumanEvalFix). A glaring observation from Figure A.10 is that all the open-weight LMs except Llama-3-70B-Inst invariably fail on getting both the classification and the corresponding edit instance right (red); especially, LMs get the classification instance wrong when they get the edit instance right in a significant number of cases (teal). These observations hint towards poor discriminative abilities in generative language models (West et al., 2023).

## 4.5 How Well does DiffBLEU Correlate with Execution-based or Static-analysis Oracles?

To understand the utility of the DiffBLEU metric and its correlation with execution-based or static-analysis based oracles, we measure the Pearson coefficients between DiffBLEU and accuracy or CodeQL scores across all the models. (1) For the subset of NoFunEdit tasks where CodeQL metric applies, the results are shown in Table 3. We observe a high correlation between Diff-BLEU and CodeQL scores for Maintainability (Pearson=0.908) and Security subsets (Pearson=0.719) averaged across different types of prompts. The correlation in the Security subset is relatively lower, possibly due to the open-ended nature of the tasks that admit multiple ways of fixing the vulnerabilities. (2) Similarly, for HumanEvalFix (not shown in Table 3), where (execution)

| Prompt | Pearson Coefficient | |
|---|---|---|
| | Maintainability | Security |
| Base | 0.9646 | 0.7401 |
| 1-Shot | 0.8766 | 0.7526 |
| CoT | 0.8239 | 0.7211 |
| CoCo | 0.9677 | 0.6638 |
| Average | 0.9080 | 0.7190 |

Table 3: Correlation between DiffBLEU and CodeQL scores.

accuracy metric applies, we observe a high Pearson coefficient of 0.978 between DiffBLEU and accuracy. Thus, DiffBLEU offers a reliable and light-weight alternative to heavy-weight execution or static-analysis based oracles.

## 5 Related Work

**Code Generation**: Prior work on evaluating code LMs has largely focused on generating functionally correct code for tasks like basic or algorithmic problem solving (Chen et al., 2021; Austin et al., 2021; Hendrycks et al., 2021; Li et al., 2022a), data science (Lai et al., 2022), Text-to-SQL (Yu et al., 2018; Li et al., 2023a), etc. HumanEval (Chen et al., 2021) serves as a widely used dataset for evaluating generative ability of LMs. HumanEval has been extended to Multipl-E (Cassano et al., 2023) for multilingual evaluation in eighteen programming languages; to HumanEval+ (Liu et al., 2023a) with more test-cases for robustness; to InstructHumanEval (CodeParrot, 2023) for instruction-following ability.

**Code Editing**: The majority of software engineering workflows involve code-editing tasks like bug fixing (Gupta et al., 2017; Muennighoff et al., 2023), performance optimizations (Madaan et al., 2023; Garg et al., 2022), improving readability and maintainability (Al Madi, 2022; Wadhwa et al., 2023; Jain et al., 2023; Loriot et al., 2022), code migration (Bairi et al., 2023), security-related edits (Perry et al., 2022; Tony et al., 2023; Pearce et al., 2022; He & Vechev, 2023; Bhatt et al., 2023; Zhuo et al., 2024), etc. More recent works like SWE-bench (Jimenez et al., 2024) and RepoCoder (Zhang et al., 2023) focus on repository-level coding tasks. SWE-bench (Jimenez et al., 2024) requires a model to generate patches (that can edit multiple files) to resolve issues like bug fixes in large Python repositories; RepoCoder (Zhang et al., 2023) constructs a code completion dataset using repository-level context under different scenarios like line, API invocation, and method body completion. CodeReviewer (Li et al., 2022b) curates a dataset of real-world code changes and associated reviewer comments aimed at automating code review related activities like change quality estimation, comment generation, and code refinement. Prior work has studied specific non-functional requirements in isolation. In contrast, with NoFunEval, we attempt to unify these requirements under a general framework of code-editing, using file-level context.

**Code Comprehension**: Prior works have considered code-comprehension tasks like clone detection (Svajlenko et al., 2014; Lu et al., 2021), defect detection (Zhou et al., 2019; Li et al., 2021; Lu et al., 2021), code explanation (Muennighoff et al., 2023; Leinonen et al., 2023), and Question-Answering over code (Liu & Wan, 2021; Lee et al., 2022; Sahu et al., 2024). Evaluating code LMs on NoFunClassify and HumanEvalClassify allowed us to directly compare and contrast their comprehension abilities with the corresponding edit abilities.

## 6 Conclusions

Code LMs are assuming an important role in the art and craft of software engineering and they should be evaluated on scenarios that matter to practitioners. We focus on two of these in this paper: ability to improve code as per non-functional requirements and ability to comprehend the relation between requirements and code semantics. Our extensive experiments found that the LMs falter on both these counts. With the rapid progress in the field, benchmarks can quickly become irrelevant. A remedy is to continuously improve the benchmarks. Our future goal is to keep extending NoFunEval by adding more languages, labeled examples spanning different requirements, and evaluation harnesses.

## 7 Reproducibility

We have open sourced the code and datasets for reproducibility of our experiments. We acknowledge that some of the code in our benchmark might have been seen by the LMs during pre-training. *However*, the LMs are unlikely to have seen the prompts constructed by us paired with the expected code revisions during pre-training. We thus observe generally poor performance on the NoFunEval Benchmark across all the LMs. Note that we do not use the actual commit messages as instructions in the prompts of our benchmark.

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

# A    Appendix

## A.1    Summary of Datasets in NoFunEval as Described in Section 2

| Task Type | Requirement | # Examples | Eval Metric |
|---|---|---|---|
| **NoFunEdit** | Latency | 47 | DiffBLEU |
| | Resource Utilization | 51 | DiffBLEU |
| | Runtime Efficiency | 113 | Average Speed-up |
| | Maintainability | 145 | DiffBLEU$\times$CodeQL |
| | Security | 41 | DiffBLEU$\times$CodeQL |
| **NoFunClassify** | All the above | 397 | Accuracy |
| **HumanEvalClassify** | Correctness | 164 | Accuracy |

Table A1: Summary of the datasets comprising the NoFunEval benchmark (§ 2).

| Task Type | Language | # Examples |
|---|---|---|
| **NoFunEdit and NoFunClassify** | Python | 277 |
| | Java | 77 |
| | C | 28 |
| | Kotlin | 4 |
| | Ino | 3 |
| | Javascript XML, Scala | 2 (each) |
| | Javascript, C++, Assembly, Objective C | 1 (each) |
| **HumanEvalClassify** | Python | 164 |

Table A2: Number of examples from different programming languages present in the NoFunEval benchmark (§ 2).

## A.2    Prompt Templates

Figure A.1: An example Base Prompt for improving bandwidth usage in code for an Android application (§ 2.2).

1-Shot/CoT Prompt. **Non-functional requirement**: Maintainability

```
Below is an instruction that describes a task. Write a response that appropriately completes the request.

### Instruction: Rewrite the given Python program to avoid the "Unguarded next in generator" CodeQL warning. Write the entire code and
no other text in the response.
      ...
def separate_headers(files):
    for file in files:
        lines = iter(file)
        header = next(lines)
        body = [l for l in lines]
        yield header, body
        ...

### Thought: The function next() will raise a StopIteration exception if the underlying iterator is exhausted, which happens in the
separate_headers() function. To avoid this, the call to next() should be wrapped in a try-except to explicitly handle StopIteration exceptions.

### Response:
      ...
def separate_headers(files):
    for file in files:
        lines = iter(file)
        try:
            header = next(lines)
        except StopIteration:
            # Empty file — Just ignore
            continue
        body = [l for l in lines]
        yield header, body
        ...
### Instruction: Rewrite the given Python program to avoid the "Unguarded next in generator" CodeQL warning. Write the entire code and
no other text in the response.
        # Input source code
### Thought:
```

Figure A.2: An example 1-Shot / Chain-of-Thought prompt template for fixing a maintainability issue ("Unguarded next in generator") as flagged by CodeQL. The underlined texts are instantiated based on the example. The shaded text denotes the reasoning we include for the corresponding Chain-of-Thought prompt (§ 2.2).

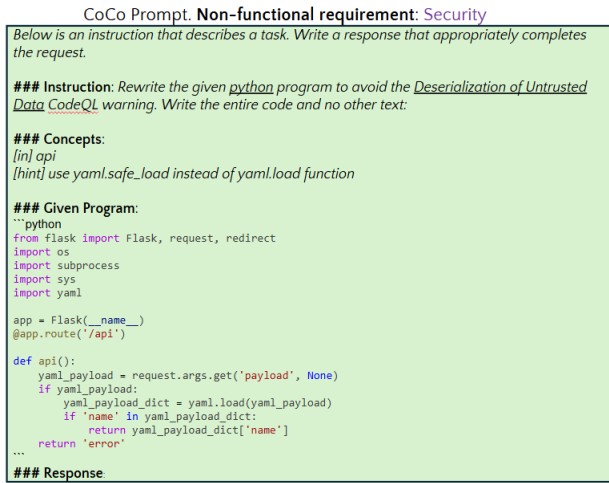

Figure A.3: An example CoCo prompt template for fixing a security issue ("Deserialization of Untrusted Data") as flagged by CodeQL (§ 2.2). The underlined texts are instantiated based on the example.

Figure A.1 provides an example base prompt used for prompting models with the high-level non-functional requirement to be achieved in editing the code. Figure A.2 highlights the template used for prompting models with a single example along with a thought relevant for the non-functional requirement being addressed. Figure A.3 highlights the template used for prompting models providing them the coding concepts needed for handling the non-functional requirement in the example shown (§ 2.2).

## A.3 Performance of Different Prompts on NoFunEdit

Figure A.4 provides a summary of all the absolute values of the evaluation numbers obtained across models and prompts. The darker shades indicating better performance on that specific task (§ 4.2).

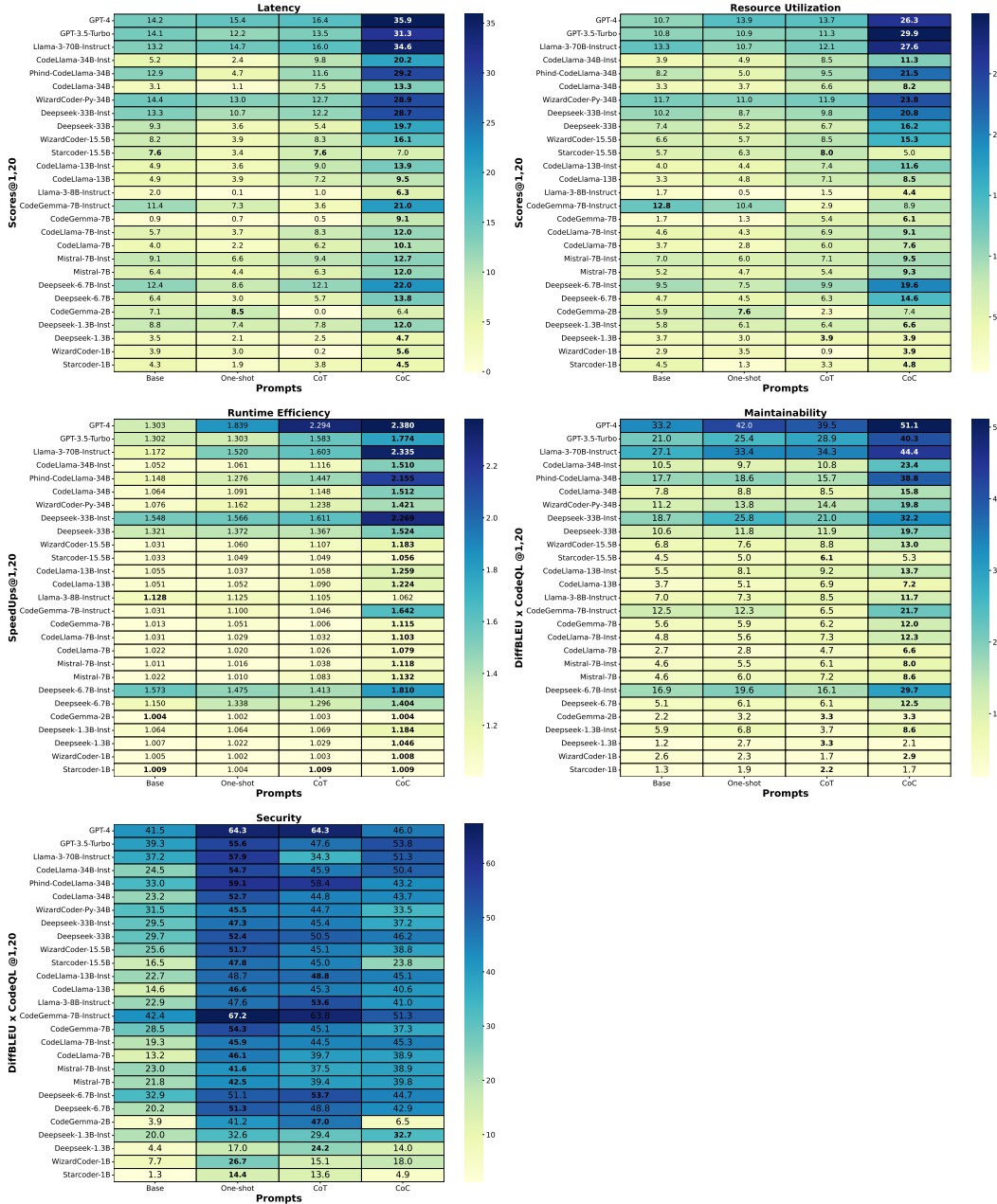

Figure A.4: Performance of code LMs on the NoFunEdit dataset by different non-functional requirements, for the four prompts (§ 4.2).

## A.4 Additional Experiments on Classification Tasks

Observing embarrassingly lower performance on classification compared to the corresponding edit tasks (§ 4.4), led us to try out more prompts to reduce the dependence of key our observations on a specific prompt format. Thus, we re-ran classification experiments with a different prompt format. In the main paper, we report numbers for the prompt that asks the model to select from "Code-A" or "Code-B" (Figure 2, R.H.S.). As an alternate, we tried prompting the model to output "Yes" or "No" using the prompt format shown in Figure A.5. This prompt takes two code snippets "Code-A" and "Code-B" as input, claims a hypothesis (e.g. Code-A is faster than Code-B), and asks the LLM to predict whether the hypothesis is correct or not by answering in "Yes" or "No". Table A3 reports numbers for NoFunClassify

| Models | Latency | Resource Utilization | RunTime Efficiency | Maintainability | Security | NoFunClassify | HumanEvalClassify (Bug Detection) |
|---|---|---|---|---|---|---|---|
| GPT-3.5-Turbo | 0.0 | **9.3** | 15.0 | 13.1 | 24.4 | 12.8 | 39.6 |
| GPT-4 | 4.1 | 3.7 | 0.9 | **49** | **95.1** | **28.9** | **90.9** |
| StarCoder-1B | 2.0 | 1.9 | 5.1 | 1.4 | 0.0 | 2.4 | 0.0 |
| WizardCoder-1B | 0.0 | 0.0 | 5.1 | 0.7 | 2.4 | 2.0 | 1.8 |
| StarCoder-15.5B | 0.0 | 0.0 | 0.0 | 2.8 | 0.0 | 1.0 | 0.6 |
| WizardCoder-15.5B | 0.0 | 0.0 | 0.0 | 8.3 | 0.0 | 3.0 | 0.0 |
| Mistral-7B | 0.0 | 1.9 | 0.0 | 3.4 | 0.0 | 1.5 | 0.0 |
| Mistral-7B-Inst | 4.1 | **9.3** | 4.2 | 6.9 | 7.3 | 6.1 | 3.0 |
| CodeLlama-7B | 0.0 | 0.0 | 0.0 | 0.7 | 0.0 | 0.3 | 0.0 |
| CodeLlama-7B-Inst | 2.0 | 1.9 | 1.7 | 0.0 | 0.0 | 1.0 | 0.6 |
| CodeLlama-13B | **18.4** | **9.3** | 4.2 | 3.4 | 0.0 | 5.8 | 0.0 |
| CodeLlama-13B-Inst | 2.0 | 7.4 | **15.3** | 4.1 | 0.0 | 7.0 | 6.1 |
| CodeLlama-34B | 2.0 | 0.0 | 0.0 | 0.0 | 0.0 | 0.2 | 4.3 |
| CodeLlama-34B-Inst | 0.0 | 1.9 | 11.9 | 4.8 | 2.4 | 5.6 | 4.3 |
| Phind-CodeLlama-34B | 6.1 | 7.4 | 10.2 | 18.6 | 9.8 | 12.4 | 9.8 |
| WizardCoder-Py-34B | 2.0 | 0.0 | 7.6 | 4.1 | 0.0 | 3.9 | 1.8 |
| DeepSeekCoder-1.3B | 8.2 | 5.6 | 0.0 | 0.0 | 0.0 | 1.7 | 1.8 |
| DeepSeekCoder-1.3B-Inst | 0.0 | 0.0 | 0.0 | 4.1 | 0.0 | 1.5 | 0.0 |
| DeepSeekCoder-6.7B | 4.1 | 0.0 | 0.0 | 0.7 | 0.0 | 0.7 | 0.0 |
| DeepSeekCoder-6.7B-Inst | 0.0 | 0.0 | 0.0 | 0.7 | 0.0 | 0.3 | 1.8 |
| DeepSeekCoder-33B | 0.0 | 0.0 | 0.0 | 0.0 | 2.4 | 0.2 | 0.0 |
| DeepSeekCoder-33B-Inst | 0.0 | 3.7 | 7.6 | 7.6 | 2.4 | 5.7 | 32.9 |
| Avg | 2.5 | 2.9 | 4.0 | 6.1 | 6.6 | 4.7 | 9.1 |
| Max | 18.4 | 9.3 | 15.3 | 49 | 95.1 | 28.9 | 90.9 |

Table A3: Accuracy of LMs (**highest** in bold; second-highest underlined) on the five non-functional requirements, entire NoFunClassify (micro-averaged over the five requirements), and HumanEvalClassify, for the alternate "Yes-No" prompts.

and HumanEvalClassify for this prompt, corresponding to Table 2. Overall, we see that "Yes" or "No" prompt in Figure A.5 leads to even lower performance for both NoFunClassify and HumanEvalClassify.

**Classification Prompt Template**

Below is an instruction that describes a task. Write a response that appropriately completes the request.

### Instruction:
Only one of the two code snippets has a lower memory usage.

**Code-A: <Source Code>**
**Code-B: <Target Code>**

Does Code-A have a lower memory usage than Code-B? Please answer in just "Yes" or "No"

### Response: "

Figure A.5: Prompt Template for alternate classification prompt in "Yes" or "No" format.

## A.5 Trade-off in Non-functional Improvements and Functional Correctness (§ 4.2)

| Model | Average SpeedUp @1,20 | Execution Accuracy @1,20 |
|---|---|---|
| **GPT-3.5-Turbo** | | |
| Base Prompt | 1.302 | 69.2 |
| One Shot | 1.303 | 79.5 |
| Chain of Thought | 1.583 | 67.6 |
| Coding Concepts | 1.774 | 48.0 |
| **GPT-4** | | |
| Base Prompt | 1.303 | 67.8 |
| One Shot | 1.839 | 69.3 |
| Chain of Thought | 2.294 | 66.0 |
| Coding Concepts | 2.380 | 59.2 |
| **WizardCoder-15.5B** | | |
| Base Prompt | 1.031 | 65.0 |
| One Shot | 1.060 | 68.1 |
| Chain of Thought | 1.107 | 27.6 |
| Coding Concepts | 1.183 | 54.8 |
| **CodeLlama-13B-Inst** | | |
| Base Prompt | 1.055 | 68.8 |
| One Shot | 1.037 | 67.0 |
| Chain of Thought | 1.058 | 45.6 |
| Coding Concepts | 1.259 | 55.9 |
| **Phind-CodeLlama-34B** | | |
| Base Prompt | 1.148 | 53.1 |
| One Shot | 1.276 | 67.3 |
| Chain of Thought | 1.447 | 37.7 |
| Coding Concepts | 2.155 | 52.7 |
| **WizardCoder-Py-34B** | | |
| Base Prompt | 1.076 | 26.9 |
| One Shot | 1.162 | 25.5 |
| Chain of Thought | 1.238 | 12.7 |
| Coding Concepts | 1.421 | 24.4 |
| **DeepSeekCoder-6.7B-Inst** | | |
| Base Prompt | 1.573 | 58.3 |
| One Shot | 1.475 | 63.8 |
| Chain of Thought | 1.413 | 45.4 |
| Coding Concepts | 1.810 | 61.7 |
| **DeepSeekCoder-33B-Inst** | | |
| Base Prompt | 1.548 | 64.4 |
| One Shot | 1.566 | 69.1 |
| Chain of Thought | 1.611 | 53.4 |
| Coding Concepts | 2.269 | 63.8 |

Table A4: Trade-off in Non-functional improvements (speed-up) and functional correctness (execution accuracy).

As discussed in Section 4.2, edits made by the model to satisfy non-functional requirements may come at the cost of functional correctness. In Table A4, we present results from a few models which compromise the functional correctness of the input code an in attempt to improve the runtime. Recall from Section 2.1, that the input code is always functionally correct (i.e., execution accuracy of 100%). Figure A.6, presents an example where GPT-4 generates an output functionally different from the input code while attempting to its improve runtime.

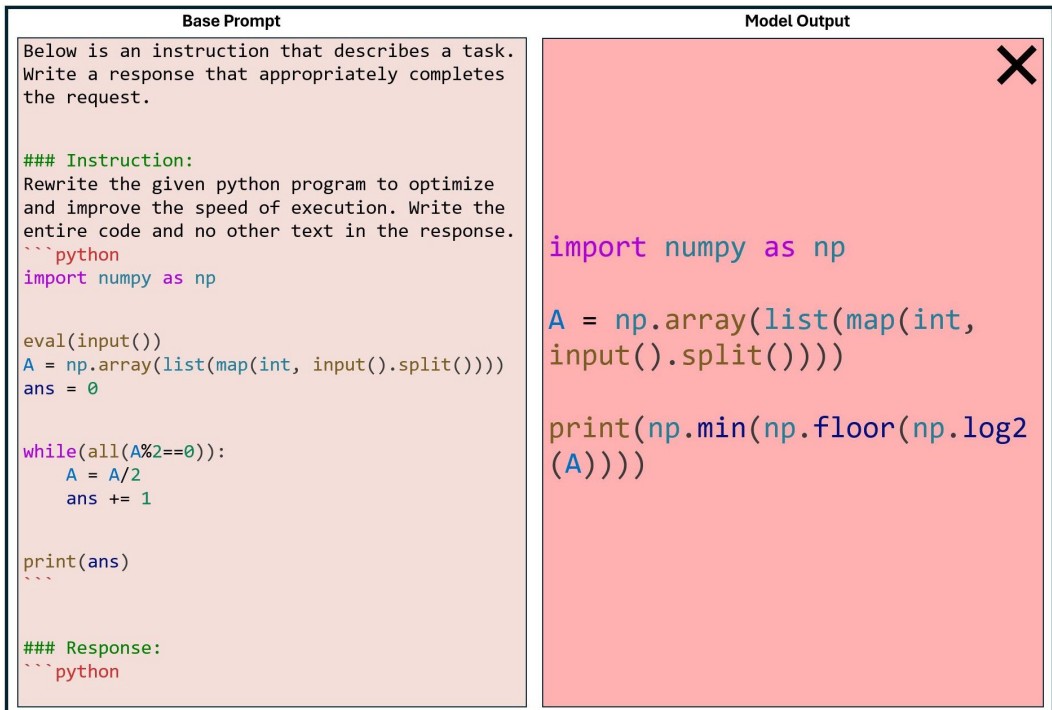

Figure A.6: In an attempt to improve the execution time, GPT-4 makes the output code functionally different from the input code, resulting in test-case violations. When all the elements of the array `A` are odd, the model-generated code (RHS) would print a non-zero number, while the original code (LHS) would print 0.

## A.6  Anecdotal Examples

Figure A.7: An example where GPT-4 utilizes hints from the CoCo prompt to successfully produce the ground truth code whereas the CoT and other prompts lead to unnecessary code edits (highlighted in green).

**1-Shot Prompt**

```
Below is an instruction that describes a task
along with an example. Write a response that
appropriately completes the request.
### Instruction:
Rewrite the given python program to optimize and
improve the speed of execution. Write the entire
code and no other text.
```python
N = int(eval(input()))
ans = 10**N - 2*9**N + 8**N
print((ans%(10**9+7)))
```

### Response:
```python
 N = int(eval(input()))
MOD = 10**9+7
ans = (pow(10, N, MOD) - 2* pow(9, N, MOD) +
pow(8, N, MOD))%MOD
print(ans)
```

### Instruction:
Rewrite the given python program to optimize and
improve the speed of execution. Write the entire
code and no other text in the response.
```python
N = int(eval(input()))
L = [ 0 ] * 1000000
L[0] = 2
L[1] = 1
for i in range(2, N+1):
  L[i] = L[i-2] + L[i-1]
print((L[N]))
```

### Response:
```python
```

**Ground Truth**

```
N = int(eval(input()))
L = [ 0 ] * (N+1)
L[0] = 2
L[1] = 1

for i in range(2, N+1):
  L[i] = L[i-2] + L[i-1]

print((L[N]))
```

**Base-Prompt Output**

```
def fibonacci(n):
    a, b = 2, 1
    for _ in range(n):
        a, b = b, a + b
    return a
N = int(input())
print(fibonacci(N))
```
✔

**1-Shot Output**

```
N = int(eval(input()))
L = [0, 1]

for _ in range(2, N+1):
    L.append(L[-2] + L[-1])

print(L[N])
```
✘

Figure A.8: Augmenting the Base prompt with 1-Shot example (1-Shot prompt) often leads to worse performance. The figure shows an example output from the GPT-4 model. Here, the Base prompt results in the correct output, even better than the ground truth (more memory efficient), however, using 1-Shot prompt (base prompt augmented with 1-Shot example) results in a functionally incorrect output, suggesting that model might learn unintended edit patterns from the 1-Shot example irrelevant to the non-functional requirement.

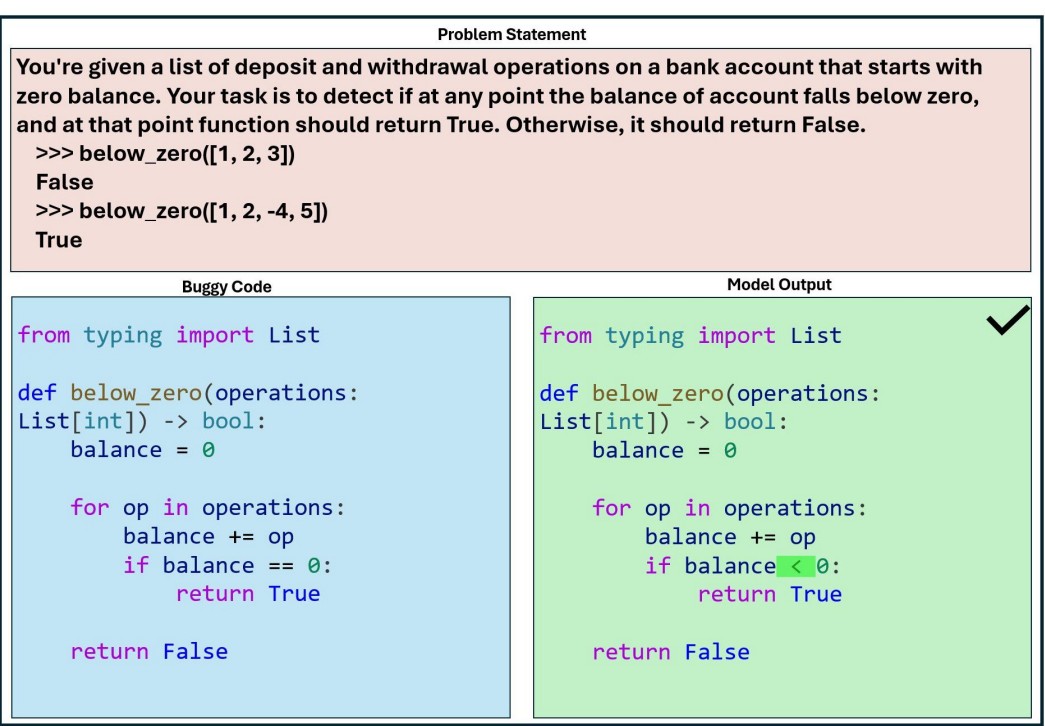

Figure A.9: An example where DeepSeekCoder-33B-Inst successfully fixes the buggy code, but fails to distinguish between the buggy and the fixed code.

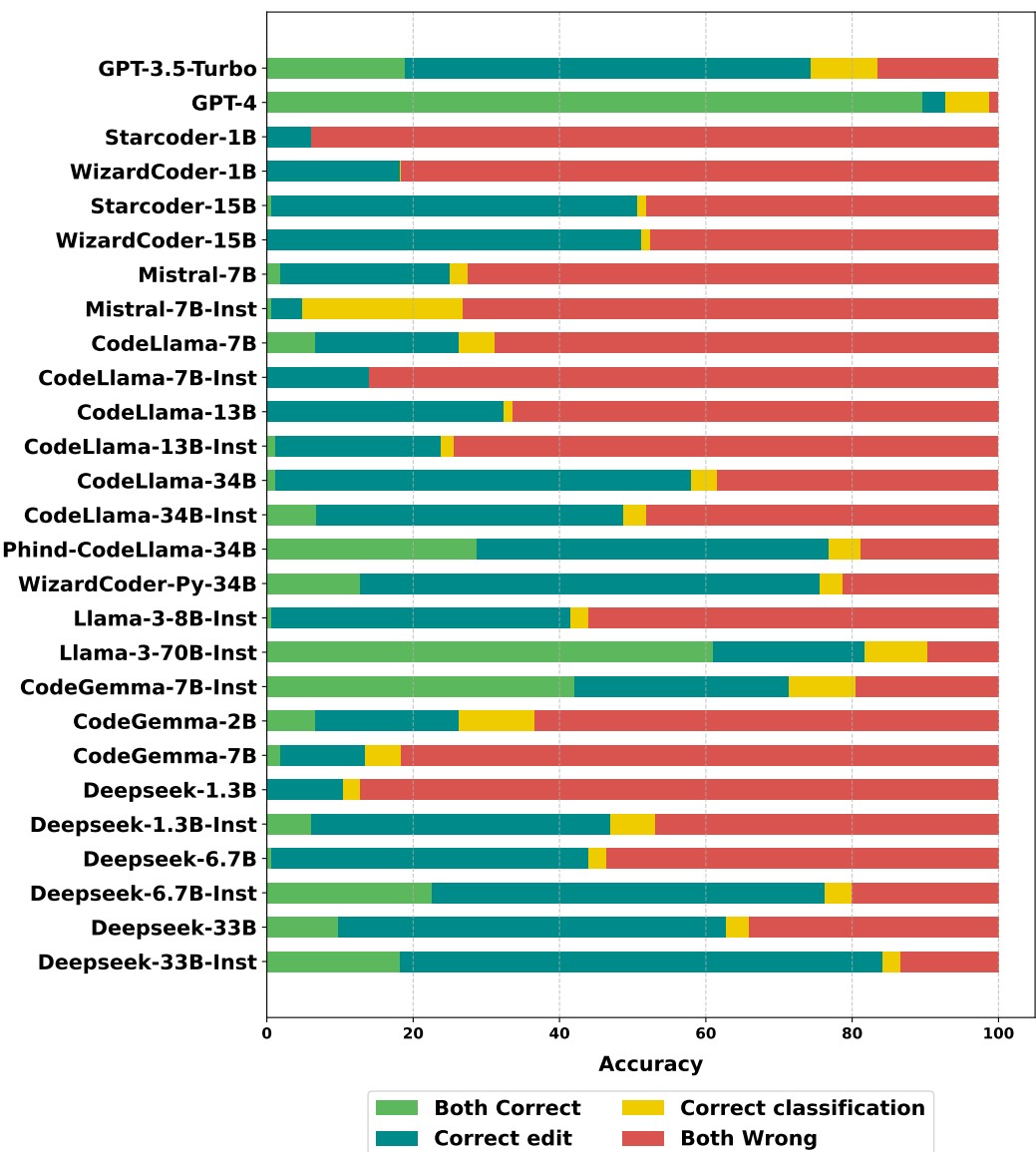

Figure A.10: Comparing code-editing (HumanEvalFix) and the corresponding code-classification (HumanEvalClassify) performance of LMs (§ 4.4). LMs fail invariably at getting *both* the classification and edit instance correct (red color). For a significant number of instances where LMs get the editing right, they fail on the classification instance (teal color).

