# OpenReview forum: "NoFunEval: Funny How Code LMs Falter on Requirements Beyond Functional Correctness"
_colmweb.org/COLM/2024/Conference — COLM_

### Official Review · Reviewer_Mb9C · 2024-05-12

**Rating:** 5
**Confidence:** 4
**Ethics Flag:** 1

**Summary:**

In this paper, the authors introduced a new benchmark called NoFunEval, focusing on code generations against other non-functional qualities of codes such as security, efficiency, etc. In addition, the authors also provide a new prompting technique called Coding Concepts (CoCo) to prompt the models with the specification of the non-functional requirements.

**Reasons To Accept:**

- The benchmark is a timely direction to address the arising research of code generation and the need to evaluate diverse qualities of codes beyond functional correctness. I found the proposed list of non-functional requirements quite comprehensive, including the code latency, resource utilization, runtime efficiency, maintainability, and security.
-Quite comprehensive evaluation with many experimental results from strong base code LLMs, including StarCoder, WizardCoder, Mistral, CodeLLama, etc. The results are fully described and analyzed with interesting insights e.g. generation vs. classification performance.

**Reasons To Reject:**

- The evaluation metrics are not ideal in some cases. For instance, in latency and resource utilization, the performance is evaluated by only DiffBLEU against the ground-truth code. It would be better to have a metric that actually measures the impacts of the generated code e.g. by real runtime, or utilized resources, even in a sandbox/ independent execution environment.
- The overall number of samples in each task is quite small, ranging from 47 to 145 samples only. Therefore, it would be quite hard to generalize the results to more general languages or coding scenarios.
- The prompting technique CoCo is rather trivial. Using the list of detailed hints/ guidelines, the models are expected to improve in their generation outputs. However, I am uncertain how these guidelines were obtained. If it was not explained in the paper, please describe how these hints/ guidelines were obtained. Are they extracted from data sources or manually written by humans?

---

> ### Author Rebuttal · Authors · 2024-05-31
>
> We thank the reviewer for the insightful review. We respond to the reviewer’s comments and questions below.
>
> > The evaluation metrics are not ideal in some cases...
>
> We use DiffBLEU exclusively for 2/5 requirements as noted by the reviewer. For others, we also perform execution- or static analysis-based evaluations.
>
> Execution-based evaluation is feasible for benchmarks like HumanEval or PIE comprising standalone functions. Our benchmark derives examples from end-to-end applications from diverse platforms like mobile devices and web applications. Setting up execution is difficult across multiple environments and is expensive to run for the benchmark users. Compared to evaluation of functional correctness using tests, measuring improvements to non-functional properties requires real-world workloads which is challenging, e.g., simulating network traffic to measure latency or replaying user requests to measure memory utilization.
>
> We hope to address these limitations in future and will discuss this in our paper. Given the high correlations between DiffBLEU and static-analysis oracles, we consider DiffBLEU as a cheaper yet reliable alternative.
>
> > The overall number of samples in each task is quite small...more general languages or coding scenarios.
>
> We agree that more examples will be helpful. But our benchmark has 397 examples, which is much higher than code-editing datasets like HumanEvalFix (164) or CanItEdit (105). Our benchmark is already multi-lingual, primarily Python, Java, and C.
>
> The raw datasets we derive NoFunEval from have orders of magnitude more examples. However, to ensure quality, we retained examples clearly consistent with non-functional requirement (e.g., filter out examples with unrelated edits - Sec 2.1). We prioritized inclusion of diverse examples covering more requirements than more but similar examples. We evaluate four different prompting strategies further adding to diversity.
>
> We aim to maintain and augment NoFunEval as a live benchmark.
>
> > The prompting technique CoCo...Are they extracted from data sources or manually written by humans?
>
> In preliminary experiments with GPT-4, we could generate CoCo prompts automatically from raw data (e.g., by summarizing ground-truth diff). Though possible to use it for benchmark creation, to ensure quality, we manually curated CoCo prompts, providing clues around what libraries to use, and what parts of code require an edit, without revealing actual edits. We will discuss this in the paper.

---

> > ### Comment · Reviewer_Mb9C · 2024-06-04
> >
> > I appreciate the authors for responding to my comments. While I noted the novelty of the proposed benchmark to test various aspects of code generation, I am still not fully convinced due to the two concerns:
> > -  The evaluation method (BLEU has been shown to correlate quite badly with functional correctness e.g. in HumanEval and MBPP papers; in Appendix A.5, the reported correlation of DiffBLEU and CodeQL scores is only for Maintainability and Security tasks); and
> > -  The benefits of the proposed CoCo prompting technique (which requires human experts to provide detailed coding concepts; this is not scalable as the concepts are quite fine-grained and require a deep understanding of the code context)

---

> > ### Author Response · Authors · 2024-06-05
> > **Thank you for acknowledging our response. We would like to add a few clarifications.**
> >
> > Thank you for acknowledging our response. We would like to clarify that:
> >
> > - In Appendix A5, we report a high correlation between DiffBLEU and Execution accuracy for HumanEvalFix dataset, in addition to correlation with CodeQL scores for Security and Maintainability. While prior work has shown BLEU to correlate poorly with functional correctness in HumanEval and MBPP benchmarks, it's important to note that unlike BLEU, DiffBLEU is specifically designed for code-edits, and that it correlates strongly with Execution accuracy for HumanEvalFix  (Pearson coefficient 0.978). We will move these details from the appendix to the main paper for more clarity in the revised version.
> >
> > - Providing additional hints via prompts like CoCo can be useful in settings where users are already familiar with the codebase they are editing. Our main objective in evaluating CoCo prompts is to understand how well different LMs can respond to additional hints.

---

### Official Review · Reviewer_mw5q · 2024-05-13

**Rating:** 6
**Confidence:** 3
**Ethics Flag:** 1

**Summary:**

The paper introduces a new benchmark, NoFunEval, to evaluate language models of code (code LMs) based on their ability to handle non-functional requirements such as efficiency, security, and maintainability, instead of simply focusing on functional correctness. The paper proposes a prompting method called Coding Concepts (CoCo) to help communication between the developers and code LMs. Twenty-two code LMs are tested using this new benchmark revealing significant shortcomings in their performance and indicating fundamental flaws in their training setups.

**Questions To Authors:**

Given various problems in the dataset, how would you efficiently generate the "concepts" for each problem? Are the concepts different from problem to problem?

**Reasons To Accept:**

The non-functional requirements are important to code LM in real applications. The experiments are extensive to show the potential direction.

**Reasons To Reject:**

The CoCo prompting needs expert-provided hints which is expensive to acquire in real scenarios.

---

> ### Author Rebuttal · Authors · 2024-05-31
>
> We  thank the reviewer for the insightful review. We respond to the reviewer’s comments and questions below.
>
> >The CoCo prompting needs expert-provided hints which is expensive to acquire in real scenarios.\
> \
> >Given various problems in the dataset, how would you efficiently generate the "concepts" for each problem? Are the concepts different from problem to problem?
>
> The purpose of the CoCo prompt is to allow a developer to communicate domain knowledge (e.g., about relevant libraries/APIs) that a model may lack. We agree that providing such hints incurs some extra effort; however, as Reviewer- Mb9C also notes, providing such hints should be trivial, particularly in the settings where the developer is familiar with the codebase they are editing. Note that at inference time, the developer has to provide hints specific to only the particular issue they are working on.
>
> Our preliminary experiments with GPT-4 suggested that we could generate these hints automatically from raw data (e.g., by summarizing the ground-truth diff) for the purposes of benchmark creation. However, in the interest of generating a high-quality dataset, we chose to manually curate CoCo prompts for each problem, mainly providing clues around what libraries to use, and what parts of code require an edit, without revealing the actual edits.

---

### Official Review · Reviewer_Qahr · 2024-05-21

**Rating:** 8
**Confidence:** 3
**Ethics Flag:** 1

**Summary:**

The authors propose and construct an evaluation suite for code language models to test their performance with respect to non-functional requirements. The authors claim to release the required code and data to the public, providing a valuable benchmark that, as the authors show, at least existing models do not perform well on.

**Questions To Authors:**

- Please include samples from every task and prompt type in the appendix

**Reasons To Accept:**

- Authors formalize an important task (non-functional requirement related performance) and construct a benchmark dataset for it
- Authors release code and dataset
- Authors show that existing models do not perform well on the task

**Reasons To Reject:**

Authors restrict evaluation to a small number of model providers (both closed and open weights). Given authors finding that GPT-4 does not perform well on the proposed benchmark, comparison to other closed-weight models such as Claude or Gemini would have been interesting.

---

> ### Author Rebuttal · Authors · 2024-05-31
>
> We thank the reviewer for the insightful review. We respond to the reviewer’s comments and questions below.
>
> >Authors restrict evaluation to a small number of model providers (both closed and open weights). Given authors finding that GPT-4 does not perform well on the proposed benchmark, comparison to other closed-weight models such as Claude or Gemini would have been interesting.
>
> Thank you for this suggestion. As presented in the paper, we have already evaluated 22 leading open and closed source code LMs of different sizes. We have since then evaluated newer open-source models like the Gemma family from Google, Llama family from Meta and DBRX from Databricks which we add below. The evaluation of these and other models is an ongoing process. We share some preliminary results below (the best prompts for the corresponding configurations are stated in parentheses). We will try to add results from these models along with Claude and Gemini in the revised version.
>
> | Models| Latency | Resource Utilization | Runtime Efficiency | Maintainability | Security |
> |-|-|-|-|-|-|
> | Meta Llama-3-8B              | 6.3 (CoCo) | 4.4 (CoCo) | 1.062 (CoCo) | 11.7 (CoCo) | 53.6 (CoT) |
> | Meta Llama-3-70B-Instruct    | 34.6 (CoCo) | 27.6 (CoCo) | 2.335 (CoCo) | 44.4 (CoCo) | 57.9 (1S) |
> | Google CodeGemma-2b          | 7.1 (Base) | 7.6 (1S) | In progress | 3.3 (CoCo) | 47.0 (CoT) |
> | Google CodeGemma-7b          | 9.1 (CoCo) | 6.1 (CoCo) | 1.115 (CoCo) | 12.0 (CoCo) | 54.3 (1S) |
> | Google CodeGemma-1.1-7b-it   | 21.0 (CoCo) | 12.8 (Base) | In progress | 21.7 (CoCo) | 67.2 (1S) |
> | Databricks DBRX-Base         | 16.0 (CoCo) | 6.7 (CoCo) | In progress | 32.4 (CoCo) | 59.2 (CoCo) |
>
> We highlight that the results on these newer models are consistent with the main observation in our paper that most models struggle to satisfy non-functional requirements, while scale in terms of model parameters and hints in the form of CoCo prompt are generally helpful.
>
> >Please include samples from every task and prompt type in the appendix
>
> Thank you for this suggestion. We will add samples from each task with examples of the different types of prompts in the appendix in the revised version. We will also be releasing the code, and all the data including the prompts publicly.

---

### Official Review · Reviewer_ddLg · 2024-05-22

**Rating:** 7
**Confidence:** 4
**Ethics Flag:** 1

**Summary:**

This paper discusses the limitations of current benchmarks for evaluating code LLMs, which primarily focus on functional correctness and use test cases to evaluate the correctness of the generated codes.  The authors propose a new benchmark called NoFunEval to conduct more code evaluations. They evaluate twenty-two code LLMs and find that they struggle on the new benchmark, indicating deficiencies in their training. Overall, this paper is well-written and clear.

**Reasons To Accept:**

1. This paper builds a new dataset to better evaluate the coding ability of LLMs.
2. This paper conducts lots of experiments on different LLMs.
3. This paper is well-motivated, well-written and clear.

**Reasons To Reject:**

1. My main concern is that there are some code benchmarks, that also aim to evaluate the coding ability of LLMs at the Non-functional level, such as SWE-bench[1], CoderReviewer[2], and RepoCoder[3]. In this case, this paper should discuss the differences between the NoFunEval and existing benchmarks, such as the new evaluation metrics, evaluation on different dimensions, and code optimization.
2. The link can not be opened.
3. The statistics of the dataset on different program languages should be shown.


[1] Can Language Models Resolve Real-world Github Issues?
[2] CodeReviewer: Pre-Training for Automating Code Review Activities.
[3] RepoCoder: Repository-Level Code Completion Through Iterative Retrieval and Generation.

---

> ### Author Rebuttal · Authors · 2024-05-31
>
> We thank the reviewer for the insightful review. We respond to the reviewer’s comments and questions below.
>
> >My main concern is that there are some code benchmarks, that also aim to evaluate the coding ability of LLMs at the Non-functional level, such as SWE-bench[1], CoderReviewer[2], and RepoCoder[3]...
>
> Thank you for pointing us to these benchmarks. As we discuss below, the objectives of these benchmarks and our work are different. We will nevertheless include this discussion in the paper.
>
> SWE-Bench and RepoCoder focus on repository-level coding tasks. SWE-Bench requires a model to generate pull-requests to resolve GitHub issues from Python repositories. These issues are primarily bug fixes, and the correctness is validated using test cases. RepoCoder constructs a dataset for code completion within repository-level context under different scenarios like line, API invocation and method body completion. These are evaluated using edit similarity and tests. Our aim in NoFunEval is distinct. We focus on evaluating whether a model can edit a given code snippet to improve a non-functional property. Due to a variety of requirements, we use corresponding oracles (like static analysis or execution, as applicable) for evaluation.
>
> CodeReviewer curates a dataset of real-world code changes and associated reviewer comments. It aims at automating code review related activities like change quality estimation, comment generation and code refinement. They measure classification accuracy and code similarity. In our work, we focus on the task of improving code as per a non-functional requirement and evaluate different LMs and prompting techniques.
>
> >The link can not be opened.
>
> We have intentionally elided the URL on page-2 to preserve anonymity. However, we have submitted our code, dataset, and results as a supplementary resource along with our submission. The URL will point to a public open-source repository in the final version of our paper.
>
> >The statistics of the dataset on different program languages should be shown.
>
> Thank you for this suggestion. The dataset primarily consists of Python, Java, and C. In addition, there are few instances in latency and resource utilization from other languages such as Kotlin, Javascript, Scala, Ino, Assembly, C++, and ObjectiveC. We will update our paper with the statistics as suggested by the reviewer.

---

> > ### Comment · Reviewer_ddLg · 2024-06-04
> > **Response to Authors**
> >
> > Even though the authors have claimed that the work is different with other benchmarks. But, I think these benchmarks also show the contribution to the No Function Eval. This paper should carefully revise to avoid the confusing points.

---

> > > ### Author Response · Authors · 2024-06-05
> > >
> > > Thank you for the follow up. We will incorporate our response in the revised version and clearly articulate the significant differences in the objectives of these benchmarks compared to ours.

---

### Decision · Program_Chairs · 2024-07-10

**Decision:**

Accept

**Comment:**

This paper proposes a new benchmark dataset for code generation, specifically on non-functional requirement related performance. This appears to be an important area in evaluating code generation models that deserves more attention. In addition, the paper also provides a comprehensive experimental study, with more results from the rebuttal. Some concerns from the reviewers, such as the exact prompting technique and more justification of the evaluation metrics, are legit and should be address properly in the next version of the paper.